# Bayesian Inference and Online Experimental Design for Mapping Neural Microcircuits

**Ben Shababo** *
Department of Biological Sciences
Columbia University, New York, NY 10027
bms2156@columbia.edu

**Brooks Paige** *
Department of Engineering Science
University of Oxford, Oxford OX1 3PJ, UK
brooks@robots.ox.ac.uk

**Ari Pakman**
Department of Statistics,
Center for Theoretical Neuroscience,
& Grossman Center for the Statistics of Mind
Columbia University, New York, NY 10027
ap3053@columbia.edu

**Liam Paninski**
Department of Statistics,
Center for Theoretical Neuroscience,
& Grossman Center for the Statistics of Mind
Columbia University, New York, NY 10027
liam@stat.columbia.edu

## Abstract

With the advent of modern stimulation techniques in neuroscience, the opportunity arises to map neuron to neuron connectivity. In this work, we develop a method for efficiently inferring posterior distributions over synaptic strengths in neural microcircuits. The input to our algorithm is data from experiments in which action potentials from putative presynaptic neurons can be evoked while a subthreshold recording is made from a single postsynaptic neuron. We present a realistic statistical model which accounts for the main sources of variability in this experiment and allows for significant prior information about the connectivity and neuronal cell types to be incorporated if available. Due to the technical challenges and sparsity of these systems, it is important to focus experimental time stimulating the neurons whose synaptic strength is most ambiguous, therefore we also develop an online optimal design algorithm for choosing which neurons to stimulate at each trial.

## 1 Introduction

A major goal of neuroscience is the mapping of neural microcircuits at the scale of hundreds to thousands of neurons [1]. By mapping, we specifically mean determining which neurons synapse onto each other and with what weight. One approach to achieving this goal involves the simultaneous stimulation and observation of populations of neurons. In this paper, we specifically address the mapping experiment in which a set of putative presynaptic neurons are optically stimulated while an electrophysiological trace is recorded from a designated postsynaptic neuron. It should be noted that the methods we present are general enough that most stimulation and subthreshold monitoring technology would be well fit by our model with only minor changes. These types of experiments have been implemented with some success [2, 3, 6], yet there are several issues which prevent efficient, large scale mapping of neural microcircuitry. For example, while it has been shown that multiple neurons can be stimulated simultaneously [4, 5], successful mapping experiments have thus far only stimulated a single neuron per trial which increases experimental time [2, 3, 6]. Stimulating multiple neurons simultaneously and with high accuracy requires well-tuned hardware, and even then some level of stimulus uncertainty may remain. In addition, a large portion of connection

---

weights are small which has meant that determining these weights is difficult and that many trials must be performed. Due to the sparsity of neural connectivity, potentially useful trials are spent on unconnected pairs instead of refining weight estimates for connected pairs when the stimuli are chosen non-adaptively. In this paper, we address these issues by developing a procedure for sparse Bayesian inference and information-based experimental design which can reconstruct neural microcircuits accurately and quickly despite the issues listed above.

## 2 A realistic model of neural microcircuits

In this section we propose a novel and thorough statistical model which is specific enough to capture most of the relevant variability in these types of experiments while being flexible enough to be used with many different hardware setups and biological preparations.

### 2.1 Stimulation

In our experimental setup, at each trial, $n = 1, \ldots, N$, the experimenter stimulates $R$ of $K$ possible presynaptic neurons. We represent the chosen set of neurons for each trial with the binary vector $\mathbf{z}_n \in \{0, 1\}^K$, which has a one in each of the the $R$ entries corresponding to the stimulated neurons on that trial. One of the difficulties of optical stimulation lies in the experimenter's inability to stimulate a specific neuron without possibly failing to stimulate the target neuron or engaging other nearby neurons. In general, this is a result of the fact that optical excitation does not stimulate a single point in space but rather has a point spread function that is dependent on the hardware and the biological tissue. To complicate matters further, each neuron has a different rheobase (a measure of how much current is needed to generate an action potential) and expression level of the optogenetic protein. While some work has shown that it may be possible to stimulate exact sets of neurons, this setup requires very specific hardware and fine tuning [4, 5]. In addition, even if a neuron fires, there is some probability that synaptic transmission will not occur. Because these events are difficult or impossible to observe, we model this uncertainty by introducing a second binary vector $\mathbf{x}_n \in \{0, 1\}^K$ denoting the neurons that actually release neurotransmitter in trial $n$. The conditional distribution of $\mathbf{x}_n$ given $\mathbf{z}_n$ can be chosen by the experimenter to match their hardware settings and understanding of synaptic transmission rates in their preparation.

### 2.2 Sparse connectivity

Numerous studies have collected data to estimate both connection probabilities and synaptic weight distributions as a function of distance and cell identity [2, 3, 6, 7, 8, 9, 10, 11, 12]. Generally, the data show that connectivity is sparse and that most synaptic weights are small with a heavy tail of strong connections. To capture the sparsity of neural connectivity, we place a "spike-and-slab" prior on the synaptic weights $w_k$ [13, 14, 15], for each presynaptic neuron $k = 1, \ldots, K$; these priors are designed to place non-zero probability on the event that a given weight $w_k$ is exactly zero. Note that we do not need to restrict the "slab" distributions (the conditional distributions of $w_k$ given that $w_k$ is nonzero) to the traditional Gaussian choice, and in fact each weight can have its own parameters. For example, log-normal [12] or exponential [8, 10] distributions may be used in conjunction with information about cell type and location to assign highly informative priors [1].

### 2.3 Postsynaptic response

In our model a subthreshold response is measured from a designated postsynaptic neuron. Here we assume the measurement is a one-dimensional trace $\mathbf{y}_n \in \mathbb{R}^T$, where $T$ is the number of samples in the trace. The postsynaptic response for each synaptic event in a given trial can be modeled using an appropriate template function $f_k(\cdot)$ for each presynaptic neuron $k$. For this paper we use an alpha function to model the shape of each neuron's contribution to the postsynaptic current, parameterized by time constants $\tau_k$ which define the rise and decay time. As with the synaptic weight priors, the template functions could be designed based on the cells' identities. The onset of each postsynaptic

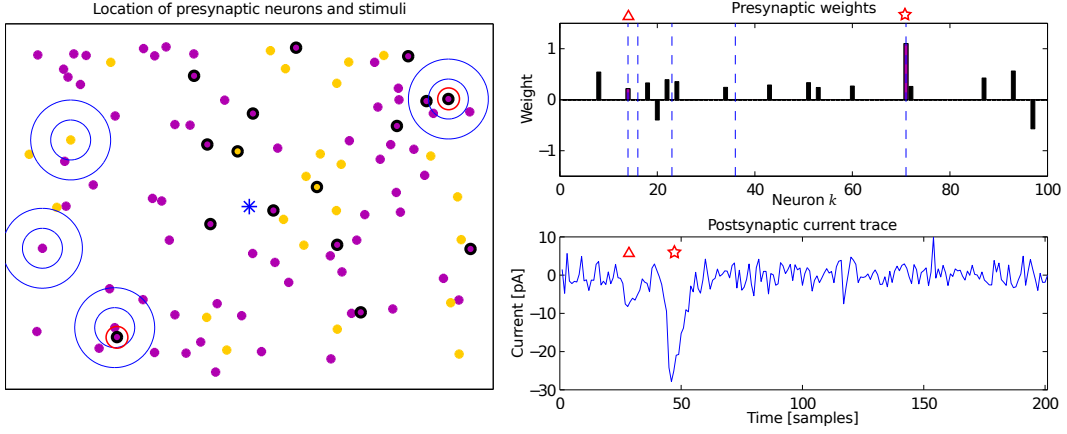

Figure 1: A schematic of the model experiment. The left figure shows the relative location of 100 presynaptic neurons; inhibitory neurons are shown in yellow, and excitatory neurons in purple. Neurons marked with a black outline have a nonzero connectivity to the postsynaptic neuron (shown as a blue star, in the center). The blue circles show the diffusion of the stimulus through the tissue. The true connectivity weights are shown on the upper right, with blue vertical lines marking the five neurons which were actually fired as a result of this stimulus. The resulting time series postsynaptic current trace is shown in the bottom right. The connected neurons which fired are circled in red, the triangle and star marking their weights and corresponding postsynaptic events in the plots at right.

response may be jittered such that each event starts at some time $d_{nk}$ after $t = 0$, where the delays could be conditionally distributed on the parameters of the stimulation and cells. Finally, at each time step the signal is corrupted by zero mean Gaussian noise with variance $\nu^2$. This noise distribution is chosen for simplicity; however, the model could easily handle time-correlated noise.

## 2.4 Full definition of model

The full model can be summarized by the likelihood

$$p(\mathbf{Y}|\mathbf{w}, \mathbf{X}, \mathbf{D}) = \prod_{n=1}^{N} \prod_{t=1}^{T} \mathcal{N}\left(y_{nt} \Big| \sum_{k} w_k x_{nk} f_k(t - d_{nk}, \tau_k), \nu^2\right) \tag{1}$$

with the general spike-and-slab prior

$$p(\gamma_k) = \text{Bernoulli}(a_k), \qquad p(w_k|\gamma_k) = \gamma_k p(w_k|\gamma_k = 1) + (1 - \gamma_k)\delta_0(w_k) \tag{2a, 2b}$$

where $\mathbf{Y} \in \mathbb{R}^{N \times T}, \mathbf{X} \in \{0, 1\}^{N \times K}$, and $\mathbf{D} \in \mathbb{R}^{N \times K}$ are composed of the responses, latent neural activity, and delays, respectively; $\gamma_k$ is a binary variable indicating whether or not neuron $k$ is connected.

We restate that the key to this model is that it captures the main sources of uncertainty in the experiment while providing room for particulars regarding hardware and the anatomy and physiology of the system to be incorporated. To infer the marginal distribution of the synaptic weights, one can use standard Bayesian methods such as Gibbs sampling or variational inference, both of which are discussed below. An example set of neurons and connectivity weights, along with the set of stimuli and postsynaptic current trace for a single trial, is shown in Figure 1.

## 3 Inference

Throughout the remainder of the paper, all simulated data is generated from the model presented above. As mentioned, any free hyperparameters or distribution choices can be chosen intelligently from empirical evidence. Biological parameters may be specific and chosen on a cell by cell basis or left general for the whole system. We show in our results that inference and optimal design still perform well when general priors are used. Details regarding data simulation as well as specific choices we make in our experiments are presented in Appendix A.

## 3.1 Charge as synaptic strength

To reduce the space over which we perform inference, we collapse the variables $w_k$ and $\tau_k$ into a single variable $c_k = \sum_t w_k f_k(t - d_{nk}, \tau_k)$ which quantifies the charge transfer during the synaptic event and can be used to define the strength of a connection. Integrating over time also eliminates any dependence on the delays $d_{nk}$. In this context, we reparameterize the likelihood as a function of $y_n = \sum_{t=0}^T y_{nt}$ and $\sigma = \nu T^{1/2}$ and the resulting likelihood is

$$p(\mathbf{y}|\mathbf{X}, \mathbf{c}) = \prod_n \mathcal{N}(y_n|\mathbf{x}_n^\top \mathbf{c}, \sigma^2). \tag{3}$$

We found that naïve MCMC sampling over the posterior of $\mathbf{w}, \boldsymbol{\tau}, \boldsymbol{\gamma}, \mathbf{X},$ and $\mathbf{D}$ insufficiently explored the support and inference was unsuccessful. In this effort to make the inference procedure computationally tractable, we discard potentially useful temporal information in the responses. An important direction for future work is to experiment with samplers that can more efficiently explore the full posterior (e.g., using Wang-Landau or simulated tempering methods).

## 3.2 Gibbs sampling

The reparameterized posterior $p(\mathbf{c}, \boldsymbol{\gamma}, \mathbf{X}|\mathbf{Z}, \mathbf{y})$ can be inferred using a simple Gibbs sampler. We approximate the prior over $\mathbf{c}$ as a spike-and-slab with Gaussian slabs where the slabs could be truncated if the cells' excitatory or inhibitory identity is known. Each $x_{nk}$ can be sampled by computing the odds ratio, and following [15] we draw each $c_k, \gamma_k$ from the joint distribution $p(c_k, \gamma_k|\mathbf{Z}, \mathbf{y}, \mathbf{X}, \{c_j, \gamma_j | j \neq k\})$ by sampling first $\gamma_k$ from $p(\gamma_k|\mathbf{Z}, \mathbf{y}, \mathbf{X}, \{c_j | j \neq k\})$, then $p(c_k|\mathbf{Z}, \mathbf{y}, \mathbf{X}, \{c_j, | j \neq k\}, \gamma_k)$.

## 3.3 Variational Bayes

As stated earlier we do not only want to recover the parameters of the system, but want to perform optimal experimental design, which is a closed-loop process. One essential aspect of the design procedure is that decisions must be returned to the experimenter quickly, on the order of a few seconds. This means that we must be able to perform inference of the posterior as well as choose the next stimulus extremely quickly. For realistically sized systems with hundred to thousands of neurons, Gibbs sampling will be too slow, and we have to explore other options for speeding up inference.

To achieve this decrease in runtime, we approximate the posterior distribution of $\mathbf{c}$ and $\boldsymbol{\gamma}$ using a variational approach [16]. The use of variational inference for spike-and-slab regression models has been explored in [17, 18], and we follow their methods with some minor changes. If we, for now, assume that $\mathbf{X}$ is known and let the spike-and-slab prior on $\mathbf{c}$ have untruncated Gaussian slabs, then this variational approach finds the best fully-factorized approximation to the true posterior

$$p(\mathbf{c}, \boldsymbol{\gamma}|\mathbf{x}_{1:n}, y_{1:n}) \approx \prod_k q(c_k, \gamma_k) \tag{4}$$

where the functional form of $q(c_k, \gamma_k)$ is itself restricted to a spike-and-slab distribution

$$q(c_k, \gamma_k) = \begin{cases} \alpha_k \mathcal{N}(c_k|\mu_k, s_k^2) & \text{if } \gamma_k = 1 \\ (1 - \alpha_k)\delta_0(c_k) & \text{otherwise.} \end{cases} \tag{5}$$

The variational parameters $\alpha_k, \mu_k, s_k$ for $k = 1, \ldots, K$ are found by minimizing the KL-divergence $KL(q||p)$ between the left and right hand sides of Eq. 4 with respect to these values. As is the case with fully-factorized variational distributions, updating the posterior involves an iterative algorithm which cycles through the parameters for each factor.

The factorized variational approximation is reasonable when the number of simultaneous stimuli, $R$, is small. Note that if we examine the posterior distributions of the weights

$$p(\mathbf{c}|\mathbf{y}, \mathbf{X}) \propto \prod_n \mathcal{N}(y_n|\mathbf{x}_n^\top \mathbf{c}, \sigma^2) \prod_k \left[ a_k \mathcal{N}(c_k|\eta_k, \sigma_k^2) + (1 - a_k)\delta_0(c_k) \right] \tag{6}$$

we see that if each $\mathbf{x}_n$ contains only one nonzero value then each factor in the likelihood is dependent on only one of the $K$ weights and can be multiplied into the corresponding $k^{th}$ spike-and-slab.

Therefore, since the product of a spike-and-slab and a Gaussian is still a spike-and-slab, if we stimulate only one neuron at each trial then this posterior is also spike-and-slab, and the variational approximation becomes exact in this limit.

Since we do not directly observe $\mathbf{X}$, we must take the expectation of the variational parameters $\alpha_k, \mu_k, s_k$ with respect to the distribution $p(\mathbf{X}|\mathbf{Z}, \mathbf{y})$. We Monte Carlo approximate this integral in a manner similar to the approach used for integrating over the hyperparameters in [17]; however, here we further approximate by sampling over potential stimuli $x_{nk}$ from $p(x_{nk} = 1|\mathbf{z}_n)$. In practice we will see this approximation suffices for experimental design, with the overall variational approach performing nearly as well for posterior weight reconstruction as Gibbs sampling from the true posterior.

## 4    Optimal experimental design

The preparations needed to perform these type of experiments tend to be short-lived, and indeed, the very act of collecting data — that is, stimulating and probing cells — can compromise the health of the preparation further. Also, one may want to use the connectivity information to perform additional experiments. Therefore it becomes critical to complete the mapping phase of the experiment as quickly as possible. We are thus strongly motivated to optimize the experimental design: to choose the optimal subset of neurons $\mathbf{z}_n$ to stimulate at each trial to minimize $N$, the overall number of trials required for good inference.

The Bayesian approach to the optimization of experimental design has been explored in [19, 20, 21]. In this paper, we maximize the mutual information $I(\boldsymbol{\theta}; \mathcal{D})$ between the model parameters $\boldsymbol{\theta}$ and the data $\mathcal{D}$; however, other objective functions could be explored. Mutual information can be decomposed into a difference of entropies, one of which does not depend on the data. Therefore the optimization reduces to the intuitive objective of minimizing the posterior entropy with respect to the data. Because the previous data $\mathcal{D}_{n-1} = \{(\mathbf{z}_1, \mathbf{y}_1), \dots, (\mathbf{z}_{n-1}, \mathbf{y}_{n-1})\}$ are fixed and $y_n$ is dependent on the stimulus $\mathbf{z}_n$, our problem is reduced to choosing the optimal next stimulus, denoted $\mathbf{z}_n^\star$, in expectation over $y_n$,

$$\mathbf{z}_n^\star = \arg\max_{\mathbf{z}_n} \mathbb{E}_{\mathbf{y}_n|\mathbf{z}_n} \left[ I(\boldsymbol{\theta}; \mathcal{D}) \right] = \arg\min_{\mathbf{z}_n} \mathbb{E}_{\mathbf{y}_n|\mathbf{z}_n} \left[ H(\boldsymbol{\theta}|\mathcal{D}) \right]. \tag{7}$$

## 5    Experimental design procedure

The optimization described in Section 4 entails performing a combinatorial optimization over $\mathbf{z}_n$, where for each $\mathbf{z}_n$ we consider an expectation over all possible $y_n$. In order to be useful to experimenters in an online setting, we must be able to choose the next stimulus in only one or two seconds. For any realistically sized system, an exact optimization is computationally infeasible; therefore in the following section we derive a fast method for approximating the objective function.

### 5.1    Computing the objective function

The variational posterior distribution of $c_k, \gamma_k$ can be used to characterize our general objective function described in Section 4. We define the cost function $J$ to be the right-hand side of Equation 7,

$$J \equiv \mathbb{E}_{y_n|\mathbf{z}_n}[H(\mathbf{c}, \boldsymbol{\gamma}|\mathcal{D})] \tag{8}$$

such that the optimal next stimulus $\mathbf{z}_n^\star$ can be found by minimizing $J$. We benefit immediately from the factorized approximation of the variational posterior, since we can rewrite the joint entropy as

$$H[\mathbf{c}, \boldsymbol{\gamma}|\mathcal{D}] \approx \sum_k H[c_k, \gamma_k|\mathcal{D}] \tag{9}$$

allowing us to optimize over the sum of the marginal entropies instead of having to compute the (intractable) entropy over the full posterior. Using the conditional entropy identity $H[c_k, \gamma_k|\mathcal{D}] = H[c_k|\gamma_k, \mathcal{D}] + H[\gamma_k|\mathcal{D}]$, we see that the entropy of each spike-and-slab is the sum of a weighted Gaussian entropy and a Bernoulli entropy and we can write out the approximate objective function as

$$J \approx \sum_k \mathbb{E}_{y_n|\mathbf{z}_n} \left[ \frac{\alpha_{k,n}}{2}(1 + \log(2\pi s_{k,n}^2)) - \alpha_{k,n} \log \alpha_{k,n} - (1 - \alpha_{k,n}) \log(1 - \alpha_{k,n}) \right]. \tag{10}$$

Here, we have introduced additional notation, using $\alpha_{k,n}$, $\mu_{k,n}$, and $s_{k,n}$ to refer to the parameters of the variational posterior distribution given the data through trial $n$. Intuitively, we see that equation 10 represents a balance between minimizing the sparsity pattern entropy $H[\gamma_k]$ of each neuron and minimizing the weight entropy $H[c_k|\gamma_k = 1]$ proportional to the probability $\alpha_k$ that the presynaptic neuron is connected. As $p(\gamma_k = 1) \rightarrow 1$, the entropy of the Gaussian slab distribution grows to dominate. In algorithm behavior, we see when the probability that a neuron is connected increases, we spend time stimulating it to reduce the uncertainty in the corresponding nonzero slab distribution.

To perform this optimization we must compute the expected joint entropy with respect to $p(y_n|\mathbf{z}_n)$. For any particular candidate $\mathbf{z}_n$, this can be Monte Carlo approximated by first sampling $y_n$ from the posterior distribution $p(y_n|\mathbf{z}_n, \mathbf{c}, \mathcal{D}_{n-1})$, where $\mathbf{c}$ is drawn from the variational posterior inferred at trial $n-1$. Each sampled $y_n$ may be used to estimate the variational parameters $\alpha_{k,n}$ and $s_{k,n}$ with which we evaluate $H[c_k, \gamma_k]$; we average over these evaluations of the entropy from each sample to compute an estimate of $J$ in Eq. 10.

Once we have chosen $\mathbf{z}_n^*$, we execute the actual trial and run the variational inference procedure on the full data to obtain the updated variational posterior parameters $\alpha_{k,n}$, $\mu_{k,n}$, and $s_{k,n}$ which are needed for optimization. Once the experiment has concluded, Gibbs sampling can be run, though we found only a limited gain when comparing Gibbs sampling to variational inference.

## 5.2   Fast optimization

The major cost to the algorithm is in the stimulus selection phase. It is not feasible to evaluate the right-hand side of equation 10 for every $\mathbf{z}_n$ because as $K$ grows there is a combinatorial explosion of possible stimuli. To avoid an exhaustive search over possible $\mathbf{z}_n$, we adopt a greedy approach for choosing which $R$ of the $K$ locations to stimulate. First we rank the $K$ neurons based on an approximation of the objective function. To do this, we propose $K$ hypothetical stimuli, $\tilde{\mathbf{z}}_n^k$, each all zeros except the $k^{\text{th}}$ entry equal to $1$ — that is, we examine only the $K$ stimuli which represent stimulating a single location. We then set $z_{nk}^* = 1$ for the $R$ neurons corresponding to the $\tilde{\mathbf{z}}_n^k$ which give the smallest values for the objective function and all other entries of $\mathbf{z}_n^*$ to zero. We found that the neurons selected by a brute force approach are most likely to be the neurons that the greedy selection process chooses (see Figure 1 in the Appendix).

For large systems of neurons, even the above is too slow to perform in an online setting. For each of the $K$ proposed stimuli $\tilde{\mathbf{z}}_n^k$, to approximate the expected entropy we must compute the variational posterior for $M$ samples of $[\mathbf{X}_{1:n-1}^\top \; \tilde{\mathbf{x}}_n^\top]^\top$ and $L$ samples of $y_n$ (where $\tilde{\mathbf{x}}_n$ is the random variable corresponding to $p(\tilde{\mathbf{x}}_n|\tilde{\mathbf{z}}_n)$). Therefore we run the variational inference procedure on the full data on the order of $\mathcal{O}(MKL)$ times at each trial. As the system size grows, running the variational inference procedure this many times becomes intractable because the number of iterations needed to converge the coordinate ascent algorithm is dependent on the correlations between the rows of $\mathbf{X}$. This is implicitly dependent on both $N$, the number of trials, and $R$, the number of stimulus locations (see Figure 2 in the Appendix). Note that the stronger dependence here is on $R$; when $R = 1$ the variational parameter updates become exact and independent across the neurons, and therefore no coordinate ascent is necessary and the runtime becomes linear in $K$.

We therefore take one last measure to speed up the optimization process by implementing an online Bayesian approach to updating the variational posterior (in the stimulus selection phase only). Since the variational posterior of $c_k$ and $\gamma_k$ takes the same form as the prior distribution, we can use the posterior from trial $n-1$ as the prior at trial $n$, allowing us to effectively summarize the previous data. In this online setting, when we stimulate only one neuron, only the parameters of that specific neuron change. If during optimization we temporarily assume that $\tilde{\mathbf{x}}_n^k = \tilde{\mathbf{z}}_n^k$, this results in explicit updates for each variational parameter, with no coordinate ascent iterations required.

In total, the resulting optimization algorithm has a runtime $\mathcal{O}(KL)$ with no coordinate ascent algorithms needed. The combined accelerations described in this section result in a speed up of several orders of magnitude which allows the full inference and optimization procedure to be run in real time, running at approximately one second per trial in our computing environment for $K = 500, R = 8$. It is worth mentioning here that there are several points at which parallelization could be implemented in the full algorithm. We chose to parallelize over $M$ which distributes the sampling of $\mathbf{X}$ and the running of variational inference for each sample. (Formulae and step-by-step implementation details are found in Appendix B.)

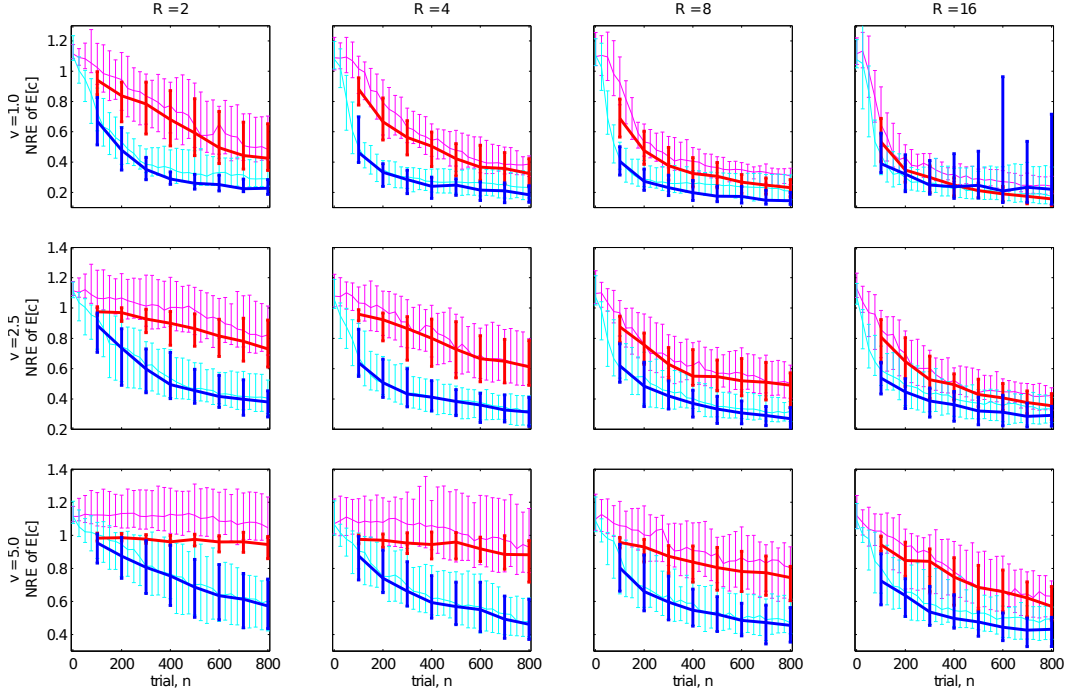

Figure 2: A comparison of normalized reconstruction error (NRE) over 800 trials in a system with 500 neurons, between random stimulus selection (red, magenta) and our optimal experimental design approach (blue, cyan). The heavy red and blue lines indicate the results when running the Gibbs sampler at that point in the experiment, and the thinner magenta and cyan lines indicate the results from variational inference. Results are shown over three noise levels $\nu = 1, 2.5, 5$, and for multiple numbers of stimulus locations per trial, $R = 2, 4, 8, 16$. Each plot shows the median and quartiles over 50 experiments. The error decreases much faster in the optimal design case, over a wide parameter range.

## 6 Experiments and results

We ran our inference and optimal experimental design algorithm on data sets generated from the model described in Section 2. We benchmarked our optimal design algorithm against a sequence of randomly chosen stimuli, measuring performance by normalized reconstruction error, defined as $\|\mathbb{E}[\mathbf{c}] - \mathbf{c}\|_2 / \|\mathbf{c}\|_2$; we report the variation in our experiments by plotting the median and quartiles.

Baseline results are shown in Figure 2, over a range of values for stimulations per trial $R$ and baseline postsynaptic noise levels $\nu$. The results here use an informative prior, where we assume the excitatory or inhibitory identity is known, and we set individual prior connectivity probabilities for each neuron based on that neuron's identity and distance from the postsynaptic cell. We choose to let $\mathbf{X}$ be unobserved and let the stimuli $\mathbf{Z}$ produce Gaussian ellipsoids which excite neurons that are located nearby. All model parameters are given in Appendix A.

We see that inference in general performs well. The optimal procedure was able to achieve equivalent reconstruction quality as a random stimulation paradigm in significantly fewer trials when the number of stimuli per trial and response noise were in an experimentally realistic range ($R = 4$ and $\nu = 2.5$ being reasonable values). Interestingly, the approximate variational inference methods performed about as well as the full Gibbs sampler here (at much less computational cost), although Gibbs sampling seems to break down when $R$ grows too large and the noise level is small, which may be a consequence of strong, local peaks in the posterior.

As the the number of stimuli per trial $R$ increases, we start to see improved weight estimates and faster convergence but a decrease in the relative benefit of optimal design; the random approach "catches up" to the optimal approach as $R$ becomes large. This is consistent with the results of [22], who argue that optimal design can provide only modest gains in performing sparse reconstructions,

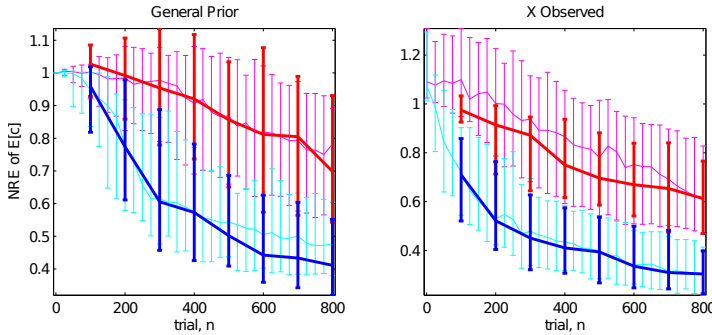

Figure 3: The results of inference and optimal design (A) with a single spike-and-slab prior for all connections (prior connection probability of .1, and each slab Gaussian with mean 0 and standard deviation 31.4); and (B) with $\mathbf{X}$ observed. Both experiments show median and quartiles range with $R = 4$ and $\nu = 2.5$.

if the design vectors $\mathbf{x}$ are unconstrained. (Note that these results do not apply directly in our setting if $R$ is small, since in this case $\mathbf{x}$ is constrained to be highly sparse — and this is exactly where we see major gains from optimal online designs.)

Finally, we see that we are still able to recover the synaptic strengths when we use a more general prior as in Figure 3A where we placed a single spike-and-slab prior across all the connections. Since we assumed the cells' identities were unknown, we used a zero-centered Gaussian for the slab and a prior connection probability of .1. While we allow for stimulus uncertainty, it will likely soon be possible to stimulate multiple neurons with high accuracy. In Figure 3B we see that - as expected - performance improves.

It is helpful to place this observation in the context of [23], which proposed a compressed-sensing algorithm to infer microcircuitry in experiments like those modeled here. The algorithms proposed by [23] are based on computing a maximum a posteriori (MAP) estimate of the weights $\mathbf{w}$; note that to pursue the optimal Bayesian experimental design methods proposed here, it is necessary to compute (or approximate) the full posterior distribution, not just the MAP estimate. (See, e.g., [24] for a related discussion.) In the simulated experiments of [23], stimulating roughly 30 of 500 neurons per trial is found to be optimal; extrapolating from Fig. 2, we would expect a limited difference between optimal and random designs in this range of $R$. That said, large values of $R$ lead to some experimental difficulties: first, stimulating large populations of neurons with high spatial resolution requires very fined tuned hardware (note that the approach of [23] has not yet been applied to experimental data, to our knowledge); second, if $R$ is sufficiently large then the postsynaptic neuron can be easily driven out of a physiologically realistic regime, which in turn means that the basic linear-Gaussian modeling assumptions used here and in [23] would need to be modified. We plan to address these issues in more depth in our future work.

## 7 Future Work

There are several improvements we would like to explore in developing this model and algorithm further. First, the implementation of an inference algorithm which performs well on the full model such that we can recover the synaptic weights, the time constants, and the delays would allow us to avoid compressing the responses to scalar values and recover more information about the system. Also, it may be necessary to improve the noise model as we currently assume that there are no spontaneous synaptic events which will confound the determination of each connection's strength. Finally, in a recent paper, [25], a simple adaptive compressive sensing algorithm was presented which challenges the results of [22]. It would be worth exploring whether their algorithm would be applicable to our problem.

### Acknowledgements

This material is based upon work supported by, or in part by, the U. S. Army Research Laboratory and the U. S. Army Research Office under contract number W911NF-12-1-0594 and an NSF CA-REER grant. We would also like to thank Rafael Yuste and Jan Hirtz for helpful discussions, and our anonymous reviewers.

## Footnotes

[1]A cell's identity can be general such as excitatory or inhibitory, or more specific such as VIP- or PV-interneurons. These identities can be identified by driving the optogenetic channel with a particular promotor unique to that cell type or by coexpressing markers for various cell types along with the optogenetic channel.

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
