[Supplementary Material]

# Bayesian Inference and Online Experimental Design for Mapping Neural Microcircuits: Supplemental Material

**Ben Shababo** [*]
Department of Biological Sciences
Columbia University, New York, NY 10027
bms2156@columbia.edu

**Brooks Paige** [*]
Department of Engineering Science
University of Oxford, Oxford OX1 3PJ, UK
brooks@robots.ox.ac.uk

**Ari Pakman**
Department of Statistics,
Center for Theoretical Neuroscience,
& Grossman Center for the Statistics of Mind
Columbia University, New York, NY 10027
ap3053@columbia.edu

**Liam Paninski**
Department of Statistics,
Center for Theoretical Neuroscience,
& Grossman Center for the Statistics of Mind
Columbia University, New York, NY 10027
liam@stat.columbia.edu

## A   Full model of neural connectivity

Here we give an extended treatment of the model introduced in Section 2, describing our specific data generation process and parameters used. This serves both to explain how we generated the data used in our experiments as well as give an example of how certain choices could be made regarding the model, e.g., the probability of neurotransmitter release given the stimuli and distributions of delays in the postsynaptic events.

### A.1   Synthetic data model

As a reminder, we model a neural microcircuit in the context of an experiment where subsets of putative presynaptic neurons are stimulated while subthreshold events are recorded from a single neuron. In our notation, we observe a postsynaptic trace $\mathbf{y}_n \in \mathbb{R}^T$ resulting from the release of neurotransmitter from a subset of $K$ putative presynaptic neurons as $\mathbf{x}_n \in \{0,1\}^K$ where $\mathbf{x}_n$ may not be observed. Each neuron influences the postsynaptic neuron with connectivity weights denoted $\mathbf{w} \in \mathbb{R}^K$ where $\mathbf{w}$ is a sparse vector representing the fact that most neurons are not synaptically connected. We also allow that each synaptic event — of which there could be multiple at each trial — has some delay $d_{nk}$, and that this delay can have a distribution which may be *iid* or may be somehow dependent on the stimuli, cells' identities, etc. For simplicity, we choose the temporal delays to be *iid* exponential distributed with mean $d_0$,

$$p(d_{nk}) = \text{Expo}(d_0). \tag{1}$$

Given these variables, we can define the likelihood

$$p(\mathbf{Y}|\mathbf{w}, \mathbf{X}, \mathbf{D}) = \prod_{n=1}^{N} \prod_{t=1}^{T} \mathcal{N}\left(y_{nt} \bigg| \sum_k w_k x_{nk} f_k(t - d_{nk}), \nu^2\right) \tag{2}$$

where $f_k(\cdot)$ is a template function which characterizes the subthreshold responses for each neuron. Again, each neurons function could be based on its identity and relationship to the postsynaptic

---

[*]These authors contributed equally to this work.

cell. Here we use an alpha-function, a common choice to characterize postsynaptic currents and potentials, which is defined as

$$f_k(t, \tau_k) = \frac{t}{\tau_k} \exp\left(-\frac{t - \tau_k}{\tau_k}\right), t \geq 0. \tag{3}$$

The constant $\tau_k$ controls the horizontal scale of the event and we draw them for each neuron from the truncated, positive normal distribution $\mathcal{N}_+(\tau_k | \mu_\tau, \sigma_\tau^2)$.

As mentioned in the main text, we use a generalized spike-and-slab prior on the weights to promote sparsity in $\mathbf{w}$. Specifically, we use lognormal distributions for the slabs [6].

$$p(w_k | \gamma_k = 1, v_k) = \ln \mathcal{N}(v_k w_k | \tilde{\eta}_k, \tilde{\sigma}_k) \tag{4}$$
$$p(w_k | \gamma_k = 0) = \delta_0(w_k) \tag{5}$$
$$p(\gamma_k | a_k) = \text{Bernoulli}(a_k) \tag{6}$$

The probability of each particular neuron $k$ having a nonzero weight, $a_k$, can be set in an informative manner based on a cell's identity and the distance from its position, $\mathbf{p}_k$, to the postsynaptic neuron at position $\mathbf{p}_0$. Nearby inhibitory neurons synapse onto excitatory neurons with a relatively high probability which falls off rather quickly as compared to excitatory-excitatory connections which occur with a lower probability at small distances and falls off more gradually as distance increases [5, 3]. Here we denote $v_k = 1$ for excitatory neurons and $v_k = -1$ for inhibitory neurons. Based on the results in [5], we use a Gaussian function to model the distribution of inhibitory to excitatory connections based on distance and an exponential for excitatory to excitatory. Note that we assume we are recording from an excitatory neuron, and also in terms of notation we use the plus and minus subscripts to identify to parameters which refer to excitatory and inhibitory neurons, respectively. This gives the prior probability of connection for a particular neuron as

$$a_k = \begin{cases} m_+ \exp(-\lambda_+ \|\mathbf{p}_k - \mathbf{p}_0\|_2), & \text{if } v_k = 1 \\ m_- \exp(-\frac{1}{2\sigma_-^2} \|\mathbf{p}_k - \mathbf{p}_0\|_2^2), & \text{if } v_k = -1 \end{cases} \tag{7}$$

A key source of uncertainty lies in the experimental inability to stimulate a particular presynaptic neuron without failing to get neurotransmitter release from that neuron or potentially stimulating nearby neurons as well. We thus treat the matrix $\mathbf{X}$ as a latent variable; we can only observe $\mathbf{Z}$, the neurons the experimenter actually indented to stimulate. The influence of stimulus $\mathbf{z}_{nr}$ is modeled as a Gaussian function of the distance from the stimulus locations to a presynaptic neuron, such that the probability a neuron at the exact stimulus location fires is at some maximum, $s_{max}$, and falls off with distance [4]. The strength of the $r^{\text{th}}$ stimulus on neuron $k$, on the $n^{\text{th}}$ trial, is given by

$$\pi_{n,k,r} = s_{max} \exp\left\{-\frac{1}{2}(\mathbf{p}_{z_{n,r}} - \mathbf{p}_k)^\top \mathbf{A}^{-1}(\mathbf{p}_{z_{n,r}} - \mathbf{p}_k)\right\} \tag{8}$$

where $A$ is a diagonal matrix that defines the spread of the stimulus through the tissue and the vectors $\mathbf{p}_{z_{n,r}}$ denote the locations of the stimuli. We then model the probability of a neuron firing on a given trial as

$$p(x_{nk} = 1 | \pi_{n,k,1...R}) = \min\left(b_{max}, \sum_{r=1}^{R} \pi_{n,k,r}\right) \tag{9}$$

in which the effect of the $R$ stimuli are additive up to a threshold. Here $b_{max} \in [0, 1]$ can be used to model the rate of synaptic transmission.

## A.2    Choosing hyperparameters

One benefit of our model is that nearly all of the hyperparameters can be chosen intelligently given available information about the hardware and previous mapping research in similar brain regions. For example, in [4] empirical distributions for firing probability given stimulus location are provided. But in general an experimenter could estimate this conditional distribution for a given setup before running the mapping procedure. Similarly, the response noise can be easily determined from the data because events are sparse. Parameters for the spike and slab prior can also be estimated by

an experimenter before hand by spending time performing initial paired patches or by consulting published results that are relevant to that particular preparation. Here we look to several papers to set the various parameters for our model [2, 3, 4, 5, 6]. In general, we assume we are recording current traces from a pyramidal neuron in somatosensory cortex of a mouse and that we use 2-photon stimulation to excite the presynaptic population.

Below is a list of hyperparameters we used for data generation:

$$\tilde{\eta}_{v_k=-1} = 1.5 \qquad \mu_\tau = 1.5 \qquad \lambda_+ = .005$$
$$\tilde{\eta}_{v_k=1} = 1 \qquad \sigma_\tau = .5 \qquad \sigma_-^2 = 5000$$
$$\tilde{\sigma}_- = 1 \qquad d_0 = 5 \qquad s_{max} = 1$$
$$\tilde{\sigma}_+ = 1 \qquad m_+ = .22 \qquad b_{max} = 1$$
$$A = \text{diag}(100, 100, 150) \qquad m_- = .5$$

When performing inference it is important to remember we are inferring $\mathbf{c}$, the charge transfer, not the amplitude of the responses $\mathbf{w}$. Thus the parameters are based on taking the integral of a mean version of the template function. In our case, that is done by using an alpha function with amplitude 1 and with a time constant $\mu_\tau$. For Gibbs sampling we use truncated Gaussians centered at zero with variances $\hat{\sigma}_+^2$ and $\hat{\sigma}_-^2$. For variational inference we shift the Gaussians so they have means $\hat{\eta}_+$ and $\hat{\eta}_-$ where

$$\hat{\eta}_+ = 58.9 \qquad \hat{\eta}_- = -78.6 \qquad \hat{\sigma}_+^2 = \hat{\sigma}_-^2 = 31.4.$$

## B  Details of optimal stimulus selection procedure

Here we present the details of the variational approximation and the optimal stimulus selection procedure. We allow sparsity, mean, and variance hyperparameters $a_k, \eta_k, \sigma_k^2$ of the spike-and-slab prior to be set individually for each of the $K$ neurons, if desired,

$$p(\mathbf{y}|\mathbf{c}, \mathbf{X}) = \prod_{n=1}^N \mathcal{N}(y_n|\mathbf{x}_n^\top \mathbf{c}, \sigma^2) \tag{10}$$

$$p(c_k|\gamma_k) = \gamma_k \mathcal{N}(c_k|\eta_k, \sigma_k^2) + (1 - \gamma_k)\delta_0(c_k) \tag{11}$$
$$p(\gamma_k = 1) = a_k. \tag{12}$$

For fixed $\mathbf{X}$, the variational approximation $\prod_k q(c_k, \gamma_k)$ to the posterior distribution is obtained via coordinate ascent on the parameters $\{\alpha_k, \mu_k, s_k\}$ using the update equations

$$s_k^2 = \frac{\sigma^2}{(\mathbf{X}^T\mathbf{X})_{kk} + \sigma^2/\sigma_k^2} \tag{13}$$

$$\mu_k = \frac{s_k^2\eta_k}{\sigma_k^2} + \frac{s_k^2}{\sigma^2}\left((\mathbf{X}^T\mathbf{y})_k - \sum_{j \neq k}(\mathbf{X}^T\mathbf{X})_{jk}\alpha_j\mu_j\right) \tag{14}$$

$$\frac{\alpha_k}{1 - \alpha_k} = \frac{a_k}{1 - a_k} \times \frac{s_k}{\sigma_k} \times \exp\left\{\frac{\mu_k^2}{2s_k^2} - \frac{\eta_k^2}{2\sigma_k^2}\right\} \tag{15}$$

which closely resemble those of [1, 7], but with additional terms to account for the fact that the slabs are not centered at zero.

In this discussion we assume that $\mathbf{X}$ is known; inference when $\mathbf{X}$ is unknown is similar, and is performed by averaging over samples from $p(\mathbf{X}|\mathbf{Z})$.

### B.1  Objective function

We use this variational posterior to select the optimal next stimulus $\mathbf{x}_n^\star$, satisfying

$$\mathbf{x}_n^\star = \underset{\mathbf{x}_n}{\arg\min}\, H[\mathbf{c}, \boldsymbol{\gamma}|\mathcal{D}] \approx \underset{\mathbf{x}_n}{\arg\min}\sum_k H[c_k, \gamma_k|\mathcal{D}]. \tag{16}$$

We define the function $\tilde{J} \equiv H[c_k, \gamma_k | \mathcal{D}]$ to be the marginal contribution of an individual neuron to the total entropy, as a function of the variational parameters

$$\tilde{J}(\alpha_{k,n}, s_{k,n}) = \frac{\alpha_{k,n}}{2}(1 + \log(2\pi s_{k,n}^2)) - \alpha_{k,n} \log \alpha_{k,n} - (1 - \alpha_{k,n}) \log(1 - \alpha_{k,n}). \quad (17)$$

To perform the optimization we must take the expectation of $\tilde{J}$ with respect to $p(y_n | \mathbf{x}_n)$. We approximate this by generating $\ell = 1, \ldots, L$ samples $y_n^{(\ell)}$ for a given $\mathbf{x}_n$. This is done by sampling $w_k, \gamma_k$ from the distribution defined by the variational posterior at trial $n - 1$,

$$p(\gamma_k^{(\ell)} | \mathcal{D}) = \text{Bernoulli}(\alpha_{k,n-1}) \quad (18)$$

$$p(c_k^{(\ell)} | \gamma_k^{(\ell)}, \mathcal{D}) = \mathcal{N}(c_k^{(\ell)} | \mu_{k,n-1}, s_{k,n-1}^2) \quad (19)$$

$$p(y_n^{(\ell)} | \mathbf{x}_n, \mathcal{D}) = \mathcal{N}(y_n^{(\ell)} | \mathbf{x}_n^\top \mathbf{c}_n^{(\ell)}, \sigma^2). \quad (20)$$

We can then use each sample $y_n^{(\ell)}$ to estimate the variational parameters $\alpha_{k,n}^{(\ell)}$ and $s_{k,n}^{(\ell)}$, yielding the finite-sample approximation

$$\mathbf{x}_n^\star = \arg\min_{\mathbf{x}_n} \sum_{k=1}^{K} \mathbb{E}\left[H[c_k, \gamma_k | \mathcal{D}]\right] \approx \arg\min_{\mathbf{x}_n} \frac{1}{L} \sum_{\ell=1}^{L} \sum_{k=1}^{K} \tilde{J}\left(\alpha_{k,n}^{(\ell)}, s_{k,n}^{(\ell)}\right). \quad (21)$$

## B.2 Stimulus selection

In our greedy selection process, we are considering those $\mathbf{x}_n^k$ in which only the $k^{\text{th}}$ entry is nonzero. In this case the updates can be further simplified, resulting in computations in which no coordinate ascent is necessary:

$$s_{k,n}^2 \approx \frac{\sigma^2 s_{k,n-1}^2}{\sigma^2 + s_{k,n-1}^2} \quad (22)$$

$$\mu_{k,n} \approx \frac{\sigma^2 \mu_{k,n-1}}{\sigma^2 + s_{k,n-1}^2} + \frac{s_{k,n}^2 y_n}{\sigma^2} \quad (23)$$

$$\frac{\alpha_{k,n}}{1 - \alpha_{k,n}} \approx \frac{\pi_k}{1 - \pi_k} \times \frac{s_{k,n}}{s_{k,n-1}} \times \exp\left\{\frac{\mu_{k,n}^2}{2s_{k,n}^2} - \frac{\mu_{k,n-1}^2}{2s_{k,n-1}^2}\right\}. \quad (24)$$

The full stimulus selection procedure is outlined in Algorithm 1 and an analysis of the runtime is presented in Figure 2.

---

**Algorithm 1** Sequential Stimuli Selection

---

Initialize $\alpha_{k,0} \leftarrow a, \mu_{k,0} \leftarrow 0, s_{k,0} \leftarrow \sigma_s$
**for** $n = 1 \rightarrow N$ **do**
    **for** $k = 1 \rightarrow K$ **do**
        $j_{(n-1)}(k) \leftarrow \tilde{J}(\alpha_{k,n-1}, s_{k,n-1})$ (eq. 17)
        **for** $\ell = 1 \rightarrow L$ **do**
            Draw sample $y_n^{(\ell)}$ (eq. 18–20)
            Compute $\alpha_{k,n}^{(\ell)}, \mu_{k,n}^{(\ell)}, s_{k,n}^{(\ell)}$ (eqn's. 22–24)
        **end for**
        $j_{(n)}(k) \leftarrow \frac{1}{L} \sum \tilde{J}(\alpha_{k,n}^{(\ell)}, s_{k,n}^{(\ell)})$
        $\Delta j(k) \leftarrow j_{(n)}(k) - j_{(n-1)}(k)$
    **end for**
    Sort $\Delta j$ descending
    Choose next $\mathbf{z}_n^\star$ to stimulate neurons corresponding to $R$ largest values of $\Delta j$
    Execute $\mathbf{z}_n^\star$ and observe a response $y_n$
    Update $\alpha_{k,n}, \mu_{k,n}, s_{k,n}$ from full data $\mathbf{x}_{1:n}, y_{1:n}$ (eqn's. 13–15)
**end for**

---

Figure 1: A comparison of our greedy stimulus selection with an exhaustive search on a small system with 20 neurons with 2 locations stimulated at each trial ($K = 20$, $R = 2$). This histogram shows the empirical distribution of the ranks of neurons selected by exhaustive search. The neurons here are sorted by the expected change in entropy, with entries on the left reducing the entropy most. The greedy approach would select the leftmost $R$ neurons; we see that the brute-force approach tends to select these neurons preferentially as well.

Figure 2: Comparison of per trial runtimes for the combined stimulus selection and inference procedure across different values for $R$, the number of stimuli per trial. The stimulus selection phase of the algorithm runs in $O(K)$ and is constant with respect to $R$ and $n$, the amount of data up to the current trial. The dependence on $R$ seen in this plot is a result of the variational inference procedure. As mentioned previously, this is a coordinate ascent algorithm whose time until convergence is dependent on the correlations in the rows of $\mathbf{X}$. As $R$ increases, this correlation increases and we see longer runtimes; the algorithm still performs fast enough to be used online under these conditions.