[Reviews · NeurIPS 2013]

Submitted by Assigned_Reviewer_3

This paper addresses the challenge of inferring synaptic connectivity in settings where one or more putative presynaptic neurons are stimulated while the membrane potential of a single postsynaptic neuron is recorded, as in modern two-photon microscopy experiments. The authors present a new probabilistic model for this experimental setup, along with a variational inference algorithm. They then develop an online, active learning algorithm to select the most informative stimuli. The efficacy of their algorithm is demonstrated using synthetic data generated from the model.

The paper presents a novel and well-motivated modeling framework for a pertinent experimental paradigm, along with an interesting online experimental design algorithm. This problem is both relevant and interesting to the NIPS community. The model that has been introduced seems quite reasonable. It is a novel contribution, and indeed applies to many experimental settings. The experimental results with synthetic data (from a more complex model) suggest that the inference algorithm is robust, and provide compelling evidence for the benefits of experimental design in the small-$R$ case.

Obviously an application to biological recordings would make a more compelling case, though this seems impractical given the space limitations and the model details that must be conveyed. Both inference and optimal experimental design require significant variational approximations and Monte Carlo estimates. The experimental results, even under the misspecified model, are encouraging, and suggest that these approximations are not too detrimental. Nonetheless, an analysis of the failure modes (e.g. sensitivity to $M$ and $L$) would be desirable. Similarly, a comparison to Gibbs sampling could also justify the variational approximations.

To the best of my knowledge, this problem has not been addressed in a Bayesian manner. The model and algorithms presented in this paper are novel and should be interesting and relevant to a computational neuroscientists within the NIPS community.
Summary: The authors provide a novel probabilistic model, inference algorithm, and optimal experimental design procedure for the interesting and relevant task of inferring synaptic connectivity given access to subthreshold membrane potential recordings and single-neuron stimulation capabilities. The result is polished and would be a nice contribution to NIPS.

Submitted by Assigned_Reviewer_6

Targeted stimulation of multiple pre-synaptic cells combined with intra-cellular recording of a single post-synaptic cell is a popular and important method for estimating synaptic connectivity between these cells. The main drawback of this approach is that it is technically difficult, so that only small number of connections can be estimated in a given experiment. The authors aim to provide statistical methods for speeding up these experiments, by firstly providing a full statistical model which takes the sparsity of neural coupling into account, and secondly by providing computationally efficient methods for choosing target-locations for stimulation.

The paper is sound and clearly explained, and the methods are evaluated using synthetic data. The drawbacks of this submission are that a) no results on real data are presented b) the neuron/noise model used is a bit simplistic (see details below) and that it is therefore not quite clear how well this method will work in a real experimental context.

Overall, my verdict is that this is a neat (but not outstanding) submission which could be a nice addition to NIPS.

Quality: The statistical model is reasonable and the evaluation of the model on synthetic data is sound. However, there are a number of properties of the model which might provide problematic on real data, and therefore, the important test of the relevance of this study will be how well it works in such a setting. In the absence of real data, it would have been important to include simulation results in a more realistic setting, i.e. one in which the model does not perfectly describe the data (e.g. using a different neuron model, correlated noise, possibly input from other neurons, feedback couplings, etc..)

Hyper-parameters: The statistical model includes some hyper-parameters which probably strongly affect model performance, but which are not set by the method. While it is true that some of these parameters can be set by using known connection-probabilities, it is unclear how appropriate this will be for a given experimental application. How sensitive is the method to "wrong" hyperparameters e.g. on the spike-and-slab prior?"

Uncorrelated noise: The method assumes uncorrelated noise on the membrane voltages, and says that it is "straightforward" to extend this to correlated noise-- while I have no doubts that it is straightforward mathematically, it would have been important to see how strongly this affects performance e.g. in the simulations. I could imagine that the inference task is substantially harder if one also has to deal with correlated noise.

Simulations: In the absence of real data, it would have been important to at least include simulation results in a more realistic setting, i.e. one in which the model does not perfectly describe the data (e.g. using a different neuron model, correlated noise, possibly input from other neurons, feedback couplings, etc..). One of the results of the simulation was that the experimental design strategy does not show a great performance benefit in the high-noise regime with multiple stimulated entries-- I would expect that this is actually the most important regime?

Clarity: The paper is mostly clearly written, but takes a long "run up" which sets up a general statistical model which is then subsequently broken down to the model which is actually used by the authors. (E.g. they write that spike-and-slab priors with any distribution can be used, and later say that they consider only the Gaussian case). I would suggest just describing what was actually done to streamline the exposition."


Originality: As far as I know, this study is original. While there have been previous studies that address estimation of synaptic connectivity (from presynaptic stimulation/postsynaptic membrane mesuarements (e.g. Hu, Leonardo, Chklovskii) this is the first such study which aims to provide a "realistic" statistical model and, importantly, which addresses the question of experimental design for this task.

Significance: The impact of this study will be be stronlgy dependent on how well this method works on real data.

Summary: This neat paper describes Bayesian methods and experimental design for inferring synaptic connectivity via stimulation of pre-synaptic cells and intra-cellular recordings of a postsynaptic cell. The statistical model set up for this task is reasonable (but does have clear limitations), and the algorithmic methods seem sound-- therefore, this is a solid (but not outstanding) submission which could be of interest for the NIPS audience.





Submitted by Assigned_Reviewer_7

A sequential Bayesian experimental design method is proposed for mapping connectivity and weight strength between stimulated presynaptic and postsynaptic neurons, the cumulative response of the latter (after potential random delays) can be observed (not just the spiking, but the subthreshold voltage trace).

The authors use an interesting model, essentially sparse linear (after simplifications) with spike and slab priors. They apply a standard variational approximation from previous work, which works well for spike and slab priors. This procedure assumes that weights are independent under the posterior, which is slightly odd for the purpose of experimental design, but their improvements seem convincing (given the experimental setup).

Sequential BED for SLMs has been done before, using less damaging posterior approximations (but not with spike&slab), the work of Steinke, Seeger, Tsuda, AISTATS 2007 should be cited. I am not that convinced about the way they optimize over z_n (or better: x_n) by ranking single-neuron
stimuli, even ignoring P(x|z). But given that, the trick of scoring very efficiently is nice:
essentially no variational reoptimization is required.

In my opinion, the paper has the following weak points. First, I find it little convincing that
a single and known f(.) is used for all neurons. I also find the simplification in Sect. 4
rather drastic: why not integrate over the delays as well? This is not explicitly said, but I
hope the experimental data is at least drawn from the original model. This should be stated more
clearly, because otherwise why even mention the original model? The model in Sect. 4 is a standard sparse linear model, nothing related to neuroscience. Previous work of Paninski etal treated more realistic models, albeit not in a Bayesian way.

Second, the treatment of X is wrong. You need to sample from P(X|Z,y), the posterior, not from
the prior. Please at least state that what you are doing is at most an approximation.
An alternative to the ranking of independent single-neuron stimuli in 5.2 would be a greedy
approach, where one component after the other in x_n is selected. Of course, this would take
longer by a factor of K.

I was also puzzled about where the averaging over X went in the scoring: do you skip it there?

Finally, experimental results are a bit slim. Essentially, the method is tested on data sampled
from the model itself. I suppose that real experiments are hard to do, but at least one could somehow use real data. Otherwise, I wonder what the motivation of the authors is. Also, while the description in the supplemental material is nice and fairly complete, details are not given in the paper. It would be possible to save quite some of the text in previous sections, and then describe the experiments better (in the paper).

Also, you mentioned Gibbs sampling in the paper, but quote no results, neither there nor in the supplement. This is not useful, either quote and describe them, or remove the remark.

Summary: Sequential Bayesian experimental design in a sparse linear model, where the combinatorial search
over inputs is approximated in a simple way. Inference is done by a variational method with fully
factorized posterior (from previous work), the sparsity prior is spike&slab.
Author Feedback

Author rebuttal: We thank all of the reviewers for the time they put into our paper and their constructive feedback. We’ll address the common issues first, then respond to the unique comments of each reviewer.

Real Data
All of the reviewers were curious about applying our work to real data, and truthfully, the authors are very curious about this as well. However, our focus here is to clearly introduce our models and algorithms. While we have access to data that fit our model, we do not have data where the true connection weights are known. Therefore we would not be able to evaluate how well we perform. That said, we are currently working on applying our work to real data, and we plan to write up these results in future work.

Hyperparameters
Another main concern is how robust our algorithms are with respect to incorrect hyperparameters. As stated in the paper, nearly all of the hyperparameters can be estimated a priori by running a set of calibration experiments. Nonetheless, we agree it is important to test our algorithms when the hyperparameters which may be estimated poorly a priori are not chosen well. In the revised version of the paper, in Figure 3 we show that our inference is robust to deviations from the true the spike and slab hyperparameters. When attempting to test the sensitive of our algorithm to tau, we found it was quite sensitive which lead to the reformulation in the revised paper where we infer charge of a synaptic event as opposed to max amplitude.

Paper Organization/Two Models
Apparently the purpose of the two different models and the organization of the paper were confusing. To be clear, we first present a model in Section 2 that is complex enough to sufficiently describe the true data and flexible enough to be used under different experimental setups. We feel this model is a worthwhile contribution, and this is the model we use for data generation. We then introduce experimental design as a goal in Section 3. To execute ED in an online setting, it was necessary to develop a simplified model with a quick inference algorithm which we describe in Section 4. In short, we introduce material as it becomes necessary to our goals: full model > ED > a simplified model for fast computation. In the revised version of the paper, this organization has been restructured such that the models are now fully described in the (new) Sections 2-3, with ED introduced in Section 4.

Gibbs Sampler
We found that the VB approach typically performed as well or better than the Gibbs sampler. In contrast to an earlier draft, Gibbs sampling output is now included in all figures. Note that Gibbs sampling performance improved in the new formulation of the problem as well.

ED Performance as Noise Level and R Change
As we discuss in Section 6, we believe that the reduction in the efficacy of ED as noise and R increase is not a negative result. As we write in the paper, the noise levels which are realistic to a voltage clamp recording are in the smaller range of values we explored, and large values of R may disrupt some of our model’s assumptions and requires specialized hardware. But even if large R was appropriate, we see that the inference procedure works very well, reconstructing the signal in very few measurements. (In the large-R regime, the results of ref [24] indicate that ED and random stimulation will perform similarly.) Ultimately, our goal is quality reconstruction in few measurements whether ED is necessary or not, and we find in particular that ED helps most in the physiologically relevant regime of modest values for R and the noise level.

Reviewer 3
“The model could include multiple post-synaptic targets... this is a significant modification, but could be considered in the conclusion.”
This is definitely something to consider; however, there is currently no way to record subthreshold responses from a population of neurons. Also, it is possible that ED will not be effective when there are so many competing independent interests (each row of the connection matrix is independent given the data). In short, the inference procedure in the paper would already work on the population level as it would simply be K independent iterations of the problem, but we are not sure how this would affect ED. We would like to explore this as future work.

Reviewer 6
“In the absence of real data... possibly input from other neurons, feedback couplings, etc...”
We are currently working on addressing some of these issues such as non-evoked events, but this may be for future work.

Reviewer 7
“Sequential BED for SLMs has been done before... even ignoring P(x|z).”
The approximations we make here - both during inference and stimulus selection are for computational purposes. We enforce that a decision must be returned to the experimenter in roughly one second or less so that decision time is on the same scale as experimental time.
Regarding Steinke et al, while it’s not an exact comparison, the runtime of the algorithm in that paper is much slower than ours which for them is reasonable since their experiments take place on larger time scales. Nonetheless, thank you for bringing this paper to our attention; it should definitely be referenced here.

“Second, the treatment of X is wrong... state that what you are doing is at most an approximation.”
You are correct: the text has been updated to clarify the treatment of X and the approximation made here. 

“An alternative to the ranking... where one component after the other in x_n is selected.”
We tried this, but this procedure takes much too long to compute due most importantly to an increase in runtime/convergence rate of VB. Inference is slowed down by a large factor which grows with R due to the iterative nature of VB. While it is hard to quantify the convergence rate in terms of R, we can say empirically that it is not tractable to score this way even when R=4.